# What Makes for Robust Local News Provision? Structural Correlates of Local News Coverage for an Entire U.S. State, and Mapping Local News Using a New Method

## Sarah Stonbely

Center for Cooperative Media, School of Communication and Media, Montclair State University,
Montclair, NJ 07043, USA; stonbelys@montclair.edu

**Abstract:** This research addresses current gaps in knowledge about local news provision: it considers the method for best understanding the robustness of a local news ecosystem, and it identifies the structural features of a community that are correlated with its level of local news provision. Most local news assessments to date have used the geographic location of the news provider as a proxy for coverage; here, I use (self-reported) coverage area as the marker of local news provision, allowing a more accurate representation of the communities being served. I find that median household income, population density, and the percentage of the population that is Hispanic are positively correlated with the number of outlets that cover a municipality, and are therefore significant indicators of local news provision. I further identify certain local news providers as "local news originators," and map the number of LNOs for the 565 municipalities that make up the state of New Jersey, making this the first study to map local news provision at this level of detail for an entire state.

**Keywords:** journalism; local news; news deserts; mapping local news

## 1. Introduction

In April 2020, as the full scale of the coronavirus pandemic became clear, local journalists were designated "essential workers", under the recognition that local news and information is crucial during times of crisis (News Media Alliance 2020; NPPA n.d.). While there is evidence that trends may be shifting, local news is still the most trusted source of news compared to national (Knight Foundation and Gallup 2019). Moreover, in the places that have lost sources of local journalism, all sorts of negative ripple effects have been empirically documented (Barthel et al. 2016; Darr et al. 2018; Filla and Johnson 2010; Gao et al. 2020; Hayes and Lawless 2018, 2015; Shaker 2014; Snyder and Strömberg 2010), confirming the importance of local news and information to a community's broader civic and structural health. This follows more than a decade of declining revenue for all but a handful of local news outlets as advertising revenue has migrated to platforms, social media, and niche sites, and audiences have fragmented or turned away from news entirely (Bell et al. 2018; Palmer and Toff 2020; Prior 2007).

It is in this context that interest in news deserts has grown, as people seek to understand the impact of these trends (Abernathy 2018; Ferrier n.d.; Stites 2011). It has become increasingly clear that robust local journalism, just as access to fresh groceries and decent banking options, is strongly correlated to elements such as median household income, education level, and race (Hamilton and Morgan 2018).

One of the main concerns taking shape as the full scope of the local journalism crisis becomes clear is the impact on less affluent, rural, and Black/Latinx communities (e.g., Wahl-Jorgensen 2019). Further, conducting comparative research that allows the statistical correlation of structural characteristics with the robustness of local news provision has been difficult and rarely attempted. One exception is a 2018 study by Napoli, Weber, McCollough, and Wang that compared 100 communities. In contrast, this study compares

565 communities—making it perhaps the largest comparative study of the structural correlates of local news provision to date.

Related to the concern about news deserts is an interest in considering local news provision in terms of the ecosystem in which it occurs. Ecosystem research provides a holistic way to understand information producers, content, and audiences, while also denoting constant change and interrelatedness, two hallmarks of the digital era (Benkler 2006). Ecosystem studies take a bird's-eye view, often understanding journalism provision at a regional, state, or national level, incorporating some combination of mapping, large-scale content analyses, network analysis, and audience studies. Journalism ecosystem studies often take advantage of the affordances of digital, including large datasets, sophisticated software, and the latest data-science capabilities, aiming to make visible the complicated relationships at play in the landscape of journalism today.

By understanding where robust journalism exists (local news "oases"), and where people are *not* having their critical information needs met (local news deserts), the many organizations that work to sustain and grow local journalism can provide more targeted relief and assistance. In this analysis of one U.S. state's local journalism, I use a novel method for mapping local news provision: coverage area (as opposed to the physical location of an outlet's headquarters or newsroom), an approach I argue is a more accurate reflection of the local news landscape. Second, I identify, based on these coverage areas, the structural correlations of local news deserts for an entire U.S. state (New Jersey) and, in doing so, contribute to the comparative empirical question of which community characteristics are most closely related to local news provision.

### 1.1. Structural Factors Relating to Local News Provision

At the core of the effort to identify the factors that create the conditions for local news oases and, conversely, for news deserts is the linking of the provision of local news to structural features of communities. This is different from showing the effects of news on various outcomes—voting patterns, political accountability, etc. Here, I focus on the inverse: on a correlation between structural features of communities such as median household income, educational attainment, population density, and demographic makeup, to the level of local news provision. Each structural feature currently hypothesized to relate to local news provision is discussed below.

One of the most hypothesized and intuitive structural factors relating to local news provision is the wealth of a community. For example, Napoli et al. (2015) found in a comparison of three communities that the richest and least racially diverse town had the most and best local news by every measure: in terms of quantity, but also when it came to whether that local news was produced by a journalist in that community ("originality"), was about that community ("localism"), and addressed a critical information need. In another example, Abernathy and her team (2018) found that "in rural counties where [news]papers have closed or merged, the average poverty rate is nearly 4 percentage points higher than the national average of 12.7 percent" (p. 24), and provided several examples of hundred-year-old-plus papers closing in poorer towns in the 2010s.

One way that a lower median household income translates into less local news is through the purchasing power and consumer spending (Hamilton and Morgan 2018). When a community has less money to spend, fewer advertising dollars are targeted toward it. When local news outlets rely on advertising, as many still do (e.g., Stonbely and George 2018), fewer local outlets can survive where fewer advertising dollars are available. Related, as the business model has shifted to emphasize subscriptions and paywalls, lower-income communities who cannot afford to pay these extra expenses are less likely to be served by 21st century hyperlocals and digital-native outlets whose incomes are based in part or in whole on audience revenue (Benson 2018). It has also been suggested that one reason that lower-income communities are less likely to have a sufficient number of local journalism providers is that they have a lower likelihood of having high-income individuals willing to undertake a journalism venture requiring time and resources with little promise of return

on investment (Napoli et al. 2015). Digital-native local startups, in particular, require years of investment before profits may be seen (Buschow 2020; Powers and Zambrano 2016).

Another structural feature of communities that may affect local news provision has been referred to as the "literacy gap", which Friedland et al. (2012) define as "the skill sets necessary to effectively navigate the digital media environment to meet [one's] critical information needs" (p. 67) (see also Hargittai and Shafer 2006; Helsper and Reisdorf 2017). This includes the most obvious definition of literacy, which is knowing how to read, but it may also include knowing how to navigate the internet to find information, knowing how to troubleshoot digital issues, or understanding how to decode the often obtuse language of politicians and other elected officials who are so often quoted in the news.[1]

For example, Hamilton and Morgan (2018) found that indicators of print news consumption such as magazine and newspaper subscriptions or newsstand sales skew toward people with higher incomes, while those with incomes in the lowest brackets reported favoring daytime talk shows, cable news, and local television news (p. 2834). Kalogeropoulos and Kleis Nielsen (2018) likewise found that greater consumption of online news is clearly associated with occupations that have better pay and require a higher level of education, as are specific news consumption habits such as consuming public media and going directly to a news outlet's website (versus seeing an article through social media or an aggregator). A rough proxy for this type of literacy is educational attainment, as used here.

Another structural factor analyzed here is the character of the community in terms of rural, suburban, and urban classification, based on population density. There is a limited amount of research on the correlation between population density and local news provision. What does exist suggests that distance from the center of a media market results in less local news and a lower quality for the local news that does exist (Campante and Do 2014; Lindgren 2009; Morgan 2013). This may be due to the fact that rural communities are by definition less heavily populated, which means they will have fewer services and fewer businesses and therefore face the economic incentive problems discussed above. Moreover, rural residents have expressed the lowest amount of satisfaction with their local news, when compared to suburban or urban residents (Stonbely 2018; Usher 2021). Second, and related, urban communities, which are by definition more densely populated, are more likely to have a diverse population, and to have nonprofit organizations and philanthropists among their population who are willing and able to fund journalism ventures (e.g., Knight Foundation 2016). In this study I am using population density converted to urban/suburban/rural characterizations to address this structural variable.[2]

Another structural variable that potentially bears relation to local news provision is municipal spending. One of the main priorities of watchdog journalism has long been monitoring government spending and highlighting fiscal corruption. Studies have shown that when a community loses a source of local journalism, corruption follows (Gao et al. 2020). In addition, it is empirically well-documented that fiscal malfeasance often thrives when there is less local news coverage of officials' behavior (e.g., Campante and Do 2014; Hogen-Esch 2011). Normatively, a municipality with higher municipal spending should have a greater number of local news outlets covering it, to ensure that money is being spent responsibly. In other words, though all communities deserve fiscal prudence on the part of their administrators, communities with larger sums at play should have a higher level of scrutiny.[3] Here, I look at whether municipal spending is related to the number of local news originators that say they serve a municipality.

The final structural characteristic examined here is demographic makeup, which has been found in other studies to correlate to local news provision. For example, Morgan (2013) found that for one North Carolina community with a large Latinx population, relevant local news was difficult to come by. "Most significantly", Morgan (p. 483) found, "Latinos, who make up half of the town's population, lack a voice online and offline, which leads to a profound disconnect within the community". Napoli et al. (2018) found similarly that the greater the percentage of Hispanics/Latinos in a community, the less likely it was to have locally oriented reporting (p. 14); other studies have documented differences in usage and

consumption among racial and ethnic groups (e.g., Pew Research Center 2015). Here, I look at whether the percentage of Hispanic or African-American population in a municipality is related to a community's number of local news originators.

Other factors that have been found to be linked to local news provision but are not explored here include connection or proximity to a university (Kim et al. 2016; Napoli et al. 2018); potential support from philanthropic organizations (Kim et al. 2016); and the presence of other local media in the community (Kim et al. 2016).

### 1.2. Local News Mapping Studies and the Problem of Depth versus Scale

The geographic unit of analysis in this study is the municipality. New Jersey has 565 municipalities[4] within its 21 counties, which means that most municipalities are quite small: the average municipality is 15 square miles (NJ Department of Labor and Workforce Development 2017). Moreover, each municipality has its own governmental structure, contributing to what Anderson (2020) identifies as one of the paramount reasons for the importance of local news in the United States: that despite the fascination with national politics, power often operates at a very micro level.

However, people's lives rarely stay within municipal boundaries—their jobs, grocery stores, entertainment venues, and places of worship are often one, two, three, or more municipalities away. Therefore, the local news that is of relevance is not only about the municipality in which they live, but also about a slightly larger region. The concept of the local news ecosystem addresses this reality: that it is not just about the town in which one lives that one needs local news, but about some larger area that likely includes several towns. Of course, local news outlets have always known this, and very few have ever limited their coverage to just a single town (even most hyperlocals cover more than one town in practice). It is this lived experience that mapping by coverage area, as I do here (rather than by the location of the news outlet itself), tries to capture.

The other quandary that ecosystem studies have grappled with is combining depth and scale. Depth in ecosystem research usually includes a comprehensive census of outlets, an accounting of news flow that includes key producers and amplifiers, and an assessment of corresponding social media. Depth may also, but need not always, include understanding consumption patterns and effects. Scale means producing this knowledge for many local news ecosystems, not just a few or a handful. Obviously, performing both is the gold standard for understanding a regional, state, or national local-journalism landscape.

Ecosystem studies that aim for depth are usually case studies, because the amount of work necessary to identify outlets, account for flow, and assess social media is considerable. Ecosystem case study methods may include ethnography or manual content analysis. The Pew Research Center study (Pew Research Center 2010) of news diffusion in Baltimore, for instance, reconstructed specific local news events by manually analyzing iterations of stories as they evolved in print, broadcast and online. Anderson's (2010) work on the Francisville Four came out of many months spent in Philadelphia newsrooms and activist communities, combining network ethnography and qualitative newsroom analysis. The Pew Research Center (2015) also sought depth when it looked at the local news ecosystems of three American cities—Denver, Macon, and Sioux City—to understand how local news was transforming in the digital age. To do so, they conducted surveys of residents, content analyses, built outlet censuses, and performed social media analysis in each city. What resulted was a detailed analysis of the cities both individually and comparatively, from which they produced much insight; however, the methodology was so detailed and labor-intensive that it could never be replicated for a large number of cities. These are classic case study designs. The amount of labor involved in replicating these works at scale is outside the realm of possibility for any academic or industry study.

Aiming for scale, Napoli et al. (2018) sought to compare the quantity and quality of local news for 100 communities around the United States. Using digital methods that included scraping more than 700 local news websites, they were able to catalogue all outlets with an online presence in each community and study the extent to which each outlets'

output included original news about the community that fulfilled a critical information need. They included only outlets that were physically located within each community, leaving out any that might serve the community from elsewhere. This method produced important comparative results but sacrificed some depth and nuance because it defined the local news ecosystem according to strict geographic parameters.

Another example of a scaled study of local news ecosystems is Abernathy's (2018, 2020) project studying the closure of local newspapers. As struggling locals are acquired by non-media corporations, Abernathy found, they are being closed or turned into "ghost" papers that provide little original hard news about the communities they purport to serve. This study covers the entire United States at the county level—clearly a scaled effort—but its geographic unit of analysis and its emphasis on newspapers obscures some important realities of the local journalism landscape in the digital age. A number of existing ecosystem studies *may* be scalable, including network analyses such as Graeff et al.'s (2014) study of the diffusion of news about the killing of Trayvon Martin, or Gordon and Johnson's (2011, 2012) studies of the Chicago digital news ecosystem. Neither were scaled; however, their leveraging of computational tools to collect data programmatically opens up broader possibilities. Scalable studies, then, can offer generalization and a valuable comparative element, but often sacrifice detail and depth.

The goal of this project was to devise a methodology that somehow achieves both depth and scale. This first iteration (method described below) was manually intensive and not easily replicable; however, the hope is that future iterations can be more efficient because the general process has been identified. In the following section, I detail the method for producing a census of local news providers, mapping them, and gathering the structural variables that allowed the comparative analysis that follows. The online component of this phase includes an interactive map that is accessible at www.newsecosystems.org/njmap.

## 2. Materials and Methods

The first step toward mapping the local news outlets serving New Jersey was a "census" of local news providers—that is, compiling a list of all accessible outlets or sources providing relevant local news. Though this process sounds straightforward, it is in fact time-consuming and resource-intensive. At the outset, several decisions must be made, and terms defined. First, what is the geographic area of focus, and what are its boundaries? For New Jersey, the boundary of the state is obvious enough, but given its position between two major media markets (New York City and Philadelphia) located in adjacent states (New York and Pennsylvania, respectively), many outlets that provide local news to and about New Jersey are based across state lines, necessitating the use of lists for those states as well—tripling the number of lists that needed to be consulted.

Similarly, which geographic unit of analysis is most appropriate? As many states, New Jersey is divided into counties and smaller local units—in this case, municipalities. Other mapping projects (e.g., Abernathy 2020, 2016) have used the county as the geographic unit of analysis; however, for a state as dense as New Jersey, local issues are decided and local life is lived at the municipal level—a county is so large that for this purpose it is almost meaningless.[5] Accordingly, most local news outlets define their coverage areas by municipality.

The next step was to define what qualifies as local news, and who is therefore a local news producer. Local news here is defined as multisourced, multiperspectival, fact-checked content produced for a public audience, rather than a closed group on social media, for example (see also Napoli et al. 2015; Zelizer 2005); it is similar to what Konieczna (2018) calls public service journalism, which is "that slice of journalism that is required by democracy". This definition also aligns closely to that used by Friedland et al. (2012) when defining critical information needs: "Critical information needs of local communities are those forms of information that are necessary for citizens and community members to live safe and healthy lives; have full access to educational, employment, and business opportunities; and to fully participate in the civic and democratic lives of their communities should

they choose" (p. v). Despite myriad developments related to the digital age—including automated journalism, business model innovation, and the incorporation of big data—the standards for what qualifies as journalism have largely continued to reflect the forms and values defined in the legacy era of the mid-to-late twentieth century (e.g., Barnhurst and Nerone 2001; Carlson 2015; Hallin 1994; Usher 2017).

Who therefore qualified as a provider of local news? Under some definitions of local news, a chamber of commerce, for example, might qualify, as some provide local economic information on their websites. Likewise, a local lifestyle magazine will often include personal health or local cultural news, though often based on a press release or in advertorial format.[6] We chose not to include information-producing civic institutions or lifestyle outlets because the primary sources of information feeding them (the government in the former case, businesses and press releases in the latter) have an incentive to provide only flattering information, with the acknowledgement that they may be important information providers for some members of a community.

The local news providers studied here include those that would be recognized as "traditional" journalism outlets—newspapers, local television, and radio stations airing local news at the top and bottom of the hour. Digital-native online newspapers, many of which are owned and/or staffed by former legacy-outlet reporters, as well as qualifying magazines serving ethnic, LGBTQ+, and religious communities, if they provide regular local news as defined above about these communities or institutions, were also considered. Further, the analyses that follow the focus on "local news originators" were also considered, which are defined and discussed in detail below.

We checked for local news provision by going to an outlet's website, or from the description of the outlet on the list. (Some lists were more detailed than others; e.g., Cision sometimes includes full paragraphs describing an outlet's content.). Television and radio stations were included if their website had a dedicated "local news" tab and that tab had current affairs (i.e., not solely cultural affairs such as event listings). The two researchers working on this task corresponded frequently about gray areas and how to clarify the parameters; however, as with any semi-subjective task of this size, some error likely occurred. We built on and referenced past studies that have sought to classify outlets in a similar way (e.g., Napoli et al. 2015, 2018; Pew Research Center 2010).

To produce an inclusive database of all local news providers, we obtained access to several sources that maintain such lists. Some are available online free of charge, while some cost several hundred dollars; we used 13 lists in total (see Appendix A). For each database we made multiple searches using the state name in different fields, as outlets are not always cataloged consistently. Because of the intense blurring of New York and Philadelphia into New Jersey, we also had to search those two states for outlets that serve New Jersey. This obviously required extra work but was necessary to produce the kind of organic ecosystem boundary that is defined not by state line but by coverage area—a key innovation of the method, as discussed in more detail below.

Once the lists were obtained, each outlet was manually entered into or checked against a master list, in which basic information about each outlet was noted, such as coverage area, frequency, original medium, and owner. For most text-based outlets, the coverage area was listed either in its "About" section or in information for advertisers. For broadcast stations, our web developer was able to pull coverage contours for nearly all stations from the Federal Communications Commission's API. This step of entering outlets into the master list was time-intensive and took the better part of one year with two people working on it.

Once the list was "final" (a list such as this can only ever be a snapshot, especially in the contemporary turbulent environment), the information was formatted and given to a developer, who made an interactive online map with several layers, including the total number of outlets serving each municipality, as well as the structural variables for each municipality, which were obtained primarily from Census or state data: median household income,[7] educational attainment,[8] municipal spending,[9] and demographic makeup.[10] The website that houses the NJ map based on this data (www.newsecosystems.org/njmap) also

includes a list of the outlets that were not able to be mapped because their coverage areas were not contiguous, were too diffuse, or were topic-based (discussed in further detail below), as well as a search function that allows people to easily see the local news providers serving their community.

Finally, it is important to acknowledge the structural bias built into the lists that are available. Most of them are made for public-relations professionals and marketers, who are inherently more interested in audiences with specific demographic characteristics—usually affluence, but sometimes ideology or race/ethnicity as well. As a result, although we used 13 databases to compile our final list, it is likely that we still did not find outlets that operate outside of these parameters, including especially grassroots zines, email subscription lists, and groups on social media.

## 3. Results

Using the process described above, we identified 779 local news providers serving New Jersey. Of the 779 total outlets, 683 (88%) are physically based in New Jersey, 42 (5%) are based in Pennsylvania (primarily Philadelphia), and 35 (4.5%) are based in New York (primarily New York City; the rest are based in Delaware, Connecticut, or elsewhere). This variable shows one way in which our method is different from mapping projects that use the location of an outlet's studios or headquarters as a proxy for the audience it serves; there are nearly 100 local news outlets that count some part of New Jersey in their coverage area that are based across state lines.

The breakdown of outlets by medium is represented in Table 1. The largest percentage is still newspapers, at 40% (N = 308), but the proportion that are online is nearly equal, at 35% (N = 276). "Online" was defined as being either digital-native (i.e., an outlet was born on the internet), or a newspaper that has ceased print operation and publishes exclusively from its website. Although almost all radio and television stations have an online presence, none of them were classified as online.

**Table 1.** Outlets by medium.

|  | **Frequency** | **Percent** |
|---|---|---|
| Newspaper | 308 | 39.5 |
| Online | 276 | 35.4 |
| Television | 100 [1] | 12.8 |
| Radio | 66 | 8.5 |
| Magazine | 29 | 3.7 |
| Total | 779 | 100.0 |

[1] To gut-check the television and radio numbers we consulted Bob Papper, a distinguished educator, researcher, and journalist with more than 40 years of experience (www.bobpapper.com), who has conducted research for the Radio Television Digital News Association (RTDNA) since 1995 and is currently director of the RTDNA/Newhouse School at Syracuse University Annual Survey. He thought our radio numbers seemed a little low, until I explained that we included only those that produce local news; he thought our television numbers seemed high—he would've guessed there were 40 to 50 stations serving New Jersey rather than 100 (Papper 2020). However, our television number includes 58 municipal-access stations, and after subtracting those we have 42 local television stations, squarely within his window. The analysis below includes those 58 stations because they do likely provide the kind of local news that we are interested in, regardless of how many or how few viewers they have. The next phase of this research project is a content analysis that will look at, among other things, which towns are covered by each outlet, lending further clarity to this issue.

Because ours is the only study we are aware of that comprehensively maps all local news providers for an entire state, it is difficult to make comparisons; however, it is important to align the findings that do exist so that the research community can continue to build on each other's work. For example, the news desert mapping studies out of the University of North Carolina's Center for Innovation and Sustainability in Local Media (CISLM) (Abernathy 2020, 2016) map the number of newspapers serving every county in the United States, based on where those outlets are physically located. In New Jersey,

CISLM's map identifies two outlets serving Burlington County: *Burlington County Times* and *Central Record*. Our study, in contrast, found 21 newspapers (out of a total of 36 outlets) serving Burlington County (* = newspaper):

| | |
|---|---|
| 201 | Burlington Twp Sun, The* |
| Bergenfield Daily Voice | Central Record, The* |
| Beverly Bee | Cinnaminson Sun, The* |
| Bordentown Current* | Courier Post* |
| Burlington County Times* | Jersey Access Group: North Burlington School District |

Jewish Community Voice*
Jewish Standard*
Marlton Sun, The*
Medford Sun, The*
Moorestown Sun, The*
Mt. Laurel Sun, The*
Palmyra Sun, The*
Paramus Post, The*
Patch-Cinnaminson
Patch-Moorestown
Patch-Wyckoff
Philadelphia (magazine)
Pine Barrens Tribune*
Press of Atlantic City, The*
Record, The*
Shamong Sun, The*
South Jersey Biz
Star-Ledger, The*
Tabernacle Sun, The*
Voorhees Sun, The*
WBGO-FM
WLIW-TV
WLVT-TV
WNDT-CD
WNYC-FM
Wyckoff-Franklin Lakes Daily Voice

Without singling out a single study for scrutiny, this example shows how the method of identifying local news deserts based on coverage area rather than where an outlet is based produces a more nuanced picture. Of course, the very important caveat is that these outlets likely do not cover every municipality they list in their coverage area in the way that provides the critical information that people need in their daily lives. The truth of the amount of coverage for Burlington County and every other locality is probably somewhere between 2 and 36 outlets.[11]

The focus on newspapers in CISLM's study does, however, speak to a deeper truth about local news ecosystems that has also been shown in prior research (Morgan 2017, pp. 7–8; Napoli et al. 2018; Pew Research Center 2010), which is that newspapers are still the primary producers of local news, while radio stations, television stations, and some digital-native local news outlets are primarily amplifiers and circulators.

Therefore the focus here is on what I am calling local news originators (LNOs), which I define as those outlets most likely to produce, by their own journalists (original), news about the community it serves (local), on a topic that addresses one of eight critical information needs as defined in Friedland et al. (2012), including stories about local education, local government, and local cultural affairs.[12] Local news originators produce the stories that will, in turn, circulate through a news ecosystem with the help of amplifiers including most television and radio stations, and social media. The total number of local news originators serving New Jersey, according to this definition, was 581 (75% of the total).

*3.1. An important Methodological Implication of Analyzing and Mapping Local News Providers Based on Coverage Area*

Though the total number of local news providers serving New Jersey was 779, the total number of *mappable* outlets was 620, and the total number of *mappable local news originators* serving New Jersey was 495 (63.5% of 779). Outlets were not mappable because their coverage areas were either not contiguous, were not clearly defined, or they are topic-driven sites; i.e., if an outlet stated vaguely that its coverage area was "the greater tri-state area" (meaning New York, Connecticut, and New Jersey), or its coverage area is not contiguous ("the Filipino population in Northern New Jersey"), or if its coverage is topic-driven (environmental issues in New Jersey), the outlet was not mapped. There were 86 unmapped local news originators, including many of the ethnic outlets—whose audiences tend to be geographically dispersed—as well as online sites about statewide topics, such as *NJ Insider*, which covers politics statewide (a list of the unmapped outlets can be found at https://newsecosystems.org/njmap/).

Because analyzing the relationship between the structural characteristics of a town and the robustness of its local news provision requires being able to identify the specific municipalities served by an outlet, the rest of the findings that follow are based on the *mappable* local news originators. This means that many of the ethnic outlets providing local news are not included in the analysis that follows, therefore possibly skewing some of the results. Though the relatively large sample size somewhat mitigates this problem, it is a methodological consideration for future research.

However, there is a second complicating situation: When outlets are mapped, they are counted in each municipality they serve; thus, most outlets are counted more than once. Obviously, how many times an outlet is counted varies according to its size and coverage area; for example, a newspaper called *The Coast Star* says it covers 52 municipalities. On the other side of the spectrum, the digital-native Morristown Green is run by a former legacy journalist who lives in Morristown and says it covers only the municipality for which it is named. Therefore, the total count of local news originators for the structural analyses below, based on the number of occurrences of a local news originator serving a municipality, is 565[13]—higher than the raw total of 495 mapped LNOs, but smaller than the raw total of 620 mappable outlets.

Quantitative Findings: The Structural Correlations of Local News Provision

Using a raw count of local news originators in terms of number serving an individual municipality, the graph below indicates news deserts (0–1 outlets serving a municipality) and news oases (8+ outlets serving a municipality) (Figure 1). The results indicate a fairly regular distribution among these categories.

0–1 outlets (29 municipalities, 5.1% of total);

2–4 outlets (108 municipalities, 19.1% of total);

5–7 outlets (191 municipalities, 33.8% of total);

8–10 outlets (116 municipalities, 20.5% of total);

11+ (121 municipalities, 21.4% of total).

However, to conduct the analyses that follow, rather than using the raw counts of local news originators per municipality, I weigh them per 10,000 capita, to account for the difference in population size among municipalities. This is standard practice when conducting large comparative analyses of different structural features of communities (e.g., Napoli et al. 2017; Powell and Chaloupka 2009). For example, say a rural municipality with a sparse population in western New Jersey has five local news providers, while a dense urban municipality such as Newark also has five local news providers; to say that their level of local news provision is equal would not be accurate since the more dense municipality has many more institutions and likely more money at play than the rural community does. My analyses therefore seek to identify the structural characteristics associated with being a municipality with a weighted count of zero to two local news originators (a news desert) or a municipality with more than eight (a news oasis).[14]

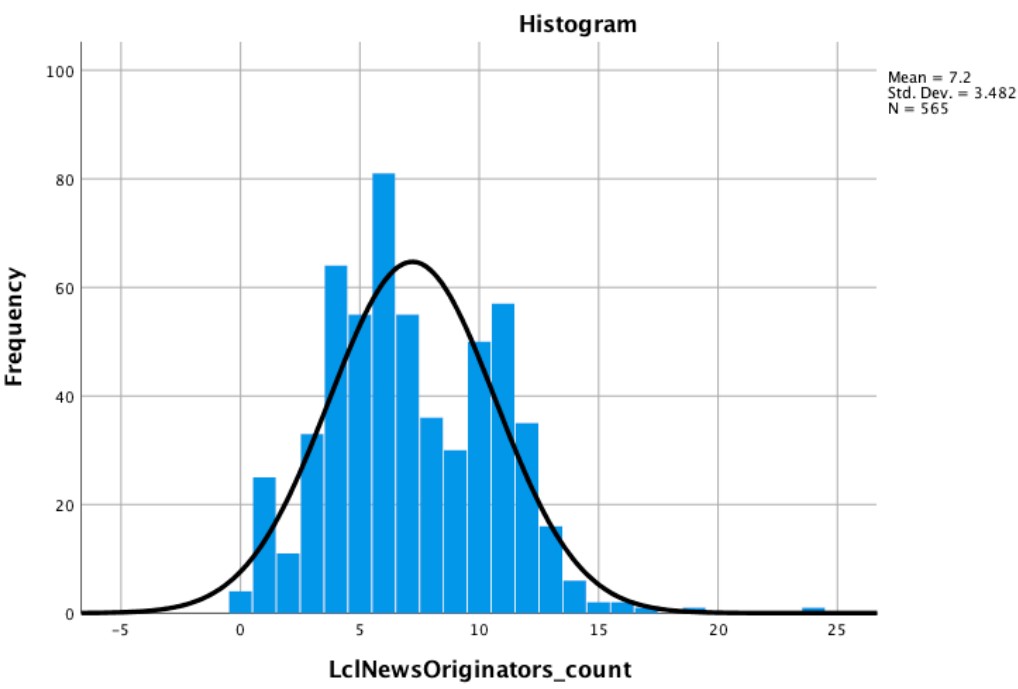

**Figure 1.** Frequency and distribution count of local news originators.

Table 2 shows the weighted count of local news originators (LNOs) serving a municipality as it relates to a municipality's median household income. The numbers in bold represent the LNO counts that are most representative in each income bracket. As one moves down the table to a higher number of local news providers, the median income that is most representative also increases, meaning that the more affluent municipalities are more likely to have a greater number of local news providers serving them. One also sees that a community in the lowest income bracket (median household income of USD 25,000 to USD 50,000) is more than twice as likely to be a news desert than a news oasis, while a community in the highest income bracket (median household income of more than USD 150,000) is extremely unlikely to be a news desert. This relationship is significant at the $p < 0.05$ level ($p = 0.054$; see Appendix C for Chi-square test results).

**Table 2.** Weighted count of local news originators per municipality by median household income.

| Weighted Count of Municipalities with Number of LNOs Serving Them | | Median Household Income (USD) | | | | | Total |
|---|---|---|---|---|---|---|---|
| | | 25–50k | 50–75k | 75–100k | 100–150k | 150k+ | |
| 0–2 LNOs per 10k pop. | Count | 14 | 39 | 22 | 11 | 1 | 87 |
| | % | 26.4% | 21.4% | 13.0% | 8.1% | 4.0% | 15.4% |
| 2–6 LNOs per 10k pop. | Count | 13 | 45 | 46 | 36 | 5 | 145 |
| | % | 24.5% | 24.7% | 27.2% | 26.5% | 20.0% | 25.7% |
| 6–12 LNOs per 10k pop. | Count | 9 | 35 | 40 | 31 | 6 | 121 |
| | % | 17.0% | 19.2% | 23.7% | 22.8% | 24.0% | 21.4% |
| 12–40 LNOs per 10k pop. | Count | 11 | 42 | 44 | 41 | 6 | 144 |
| | % | 20.8% | 23.1% | 26.0% | 30.1% | 24.0% | 25.5% |
| 40 or more LNOs per 10k pop. | Count | 6 | 21 | 17 | 17 | 7 | 68 |
| | % | 11.3% | 11.5% | 10.1% | 12.5% | 28.0% | 12.0% |
| Total | Count | 53 | 182 | 169 | 136 | 25 | 565 |
| | % | 100.0% | 100.0% | 100.0% | 100.0% | 100.0% | 100.0% |

Table 3 shows the weighted count of local news originators (LNOs) serving a municipality as it relates to a municipality's educational attainment. Here, we see again that the municipalities with the lowest educational attainment are also the most likely to be news deserts; that is, the municipalities with the lowest educational attainment have the fewest local news originators serving them, with the number increasing as education increases. However, there is not a marked pattern of increase as the level of education increases; that is, an increase in the level of education does not make a community more likely to have a greater number of local news originators. This relationship was not statistically significant (see Appendix C for Chi-square results).

**Table 3.** Number of local news originators per municipality by average educational attainment.

| Weighted Count of Municipalities with Number of LNOs Serving Them | | Educational Attainment | | | | Total |
|---|---|---|---|---|---|---|
| | | 0–30% B.A. or Higher | 30–50% B.A. or Higher | 50–70% B.A. or Higher | 70%+ B.A. or Higher | |
| 0 to 2 LNOs per 10k pop. | Count | 45 | 24 | 13 | 5 | 87 |
| | % | 51.7% | 27.6% | 14.9% | 5.7% | 100.0% |
| 2 to 6 LNOs per 10k pop. | Count | 50 | 54 | 36 | 5 | 145 |
| | % | 34.5% | 37.2% | 24.8% | 3.4% | 100.0% |
| 6 to 12 LNOs per 10k pop. | Count | 33 | 43 | 39 | 6 | 121 |
| | % | 27.3% | 35.5% | 32.2% | 5.0% | 100.0% |
| 12 to 40 LNOs per 10k pop. | Count | 45 | 57 | 33 | 9 | 144 |
| | % | 31.3% | 39.6% | 22.9% | 6.3% | 100.0% |
| 40 or more LNOs per 10k pop. | Count | 22 | 23 | 18 | 5 | 68 |
| | % | 32.4% | 33.8% | 26.5% | 7.4% | 100.0% |
| Total | Count | 195 | 201 | 139 | 30 | 565 |
| | % | 34.5% | 35.6% | 24.6% | 5.3% | 100.0% |

Table 4 shows the number of local news originators (LNOs) serving a municipality as it relates to a municipality's character (urban, suburban, rural). Because these categories are based on population counts (see footnote 2 for how the categories were constructed), it does not make sense to use the population-weighted LNO variable.

**Table 4.** Number of local news originators per municipality by character of community.

| Count of Municipalities with Number of LNOs Serving Them | | Character of Community (Pop. Density) | | | Total |
|---|---|---|---|---|---|
| | | Rural | Suburban | Urban | |
| 0–1 outlets | Count | 21 | 8 | 0 | 29 |
| | % | 72.4% | 27.6% | 0.0% | 100.0% |
| 2–4 outlets | Count | 27 | 80 | 1 | 108 |
| | % | 25.0% | 74.1% | 0.9% | 100.0% |
| 5–7 outlets | Count | 63 | 119 | 9 | 191 |
| | % | 33.0% | 62.3% | 4.7% | 100.0% |
| 8–10 outlets | Count | 8 | 93 | 15 | 116 |
| | % | 6.9% | 80.2% | 12.9% | 100.0% |
| 11+ outlets | Count | 7 | 98 | 16 | 121 |
| | % | 5.8% | 81.0% | 13.2% | 100.0% |
| Total | Count | 126 | 398 | 41 | 565 |
| | % | 22.3% | 70.4% | 7.3% | 100.0% |

Two things are apparent here. First, suburban communities are by far the highest percentage in each bracket, showing that the suburbs are the best served no matter how many local news originators we are looking at. Second, and related, the only bracket in which the rural communities score the highest is zero to one local news provider, meaning that rural communities are the most likely to be news deserts. One also notices that there are no urban communities that have zero to one outlets, and that the number of urban communities in each bracket steadily increases as the number of outlets increases, showing that population density is positively associated with a greater number of local news originators. This relationship is highly significant at the $p < 0.001$ level (see Appendix C).

Table 5 looks at the relationship between municipal spending (per 10k capita) and the weighted number of local news originators serving a municipality. The per capita number was generated by weighing municipal spending according to a municipality's population (i.e., dividing a municipality's annual budget by its population); as with LNO count, this is carried out to create a more meaningful comparative representation rather than just using raw numbers.

**Table 5.** Number of local news originators per municipality by annual municipal spending per capita.

| Weighted Count of Municipalities with Number of LNOs Serving Them | | Municipal Spending (Annual (2016/7), per 10k Capita) | | | Total |
| --- | --- | --- | --- | --- | --- |
| | | 0 to USD 1000 Spent per Capita | USD 1000 to USD 2000 Spent per Capita | >USD 3000 Spent per Capita | |
| 0–2 LNOs per 10k pop. | Count | 15 | 54 | 18 | 87 |
| | % | 17.2% | 62.1% | 20.7% | 100.0% |
| 2–6 LNOs per 10k pop. | Count | 26 | 92 | 27 | 145 |
| | % | 17.9% | 63.4% | 18.6% | 100.0% |
| 6–12 LNOs per 10k pop. | Count | 25 | 61 | 34 | 120 |
| | % | 20.8% | 50.8% | 28.3% | 100.0% |
| 12–40 LNOs per 10k pop. | Count | 31 | 65 | 46 | 142 |
| | % | 21.8% | 45.8% | 32.4% | 100.0% |
| 40 or more LNOs per 10k pop. | Count | 10 | 22 | 33 | 65 |
| | % | 15.4% | 33.8% | 50.8% | 100.0% |
| Total | Count | 107 | 294 | 158 | 559 |
| | % | 19.1% | 52.6% | 28.3% | 100.0% |

This relationship is highly significant at the $p < 0.001$ level (see Appendix C), and one can observe that the communities that spend the most money per capita also have the highest number of local news originators covering them. Normatively, this is as it should be, because one of the main functions of journalism is to watchdog how government officials spend public money. The middle bracket of municipal spending is the largest, covering more than half of the municipalities; the amount of coverage these communities receive varies widely. Notably, there are 18 communities that spend more than USD 3000 per capita and 54 communities spending between USD 1000 and USD 2000 per capita that have a weighted count of between zero and two local news originators per capita covering them. I will discuss this in more detail below.

Finally, Tables 6 and 7 show the relationship between a community's racial and ethnic makeup and its local news provision. A total of 76 of the state's 565 municipalities are "majority-minority", meaning that a population other than White makes up more than 50% of the total (Wu 2011). In five New Jersey municipalities, minorities comprise more than 90% of the population (they are Lawnside Borough, Camden City, Orange Township, Plainfield in Union County, and Paterson); non-Hispanic whites comprise more than 90% of the population in 137 municipalities (ibid.). As of the 2010 Census, New Jersey had

the nation's seventh-largest Hispanic population (Wu 2011), with 20.9% of the state's total population identifying in that way (U.S. Census Bureau 2020), making it New Jersey's largest minority community.[15] African-Americans make up 15.1% of the population, with some number of African-Americans living in all but six small municipalities.[16]

**Table 6.** Number of local news originators per municipality by percentage of population that is Hispanic.

| Weighted Count of Municipalities with Number of LNOs Serving Them | | Percent of Muni. pop. That Is Hispanic | | | Total |
|---|---|---|---|---|---|
| | | 0 to 5% of pop. Is Hispanic | 5% to 10% of pop. Is Hispanic | >10% of pop. Is HISPANIC | |
| 0 to 2 LNOs per 10k pop. | Count | 16 | 31 | 40 | 87 |
| | % | 18.4% | 35.6% | 46.0% | 100.0% |
| 2 to 6 LNOs per 10k pop. | Count | 43 | 67 | 35 | 145 |
| | % | 29.7% | 46.2% | 24.1% | 100.0% |
| 6 to 12 LNOs per 10k pop. | Count | 46 | 34 | 41 | 121 |
| | % | 38.0% | 28.1% | 33.9% | 100.0% |
| 12 to 40 LNOs per 10k pop. | Count | 52 | 46 | 46 | 144 |
| | % | 36.1% | 31.9% | 31.9% | 100.0% |
| 40 or more LNOs per 10k pop. | Count | 34 | 15 | 19 | 68 |
| | % | 50.0% | 22.1% | 27.9% | 100.0% |
| Total | Count | 191 | 193 | 181 | 565 |
| | % | 33.8% | 34.2% | 32.0% | 100.0% |

**Table 7.** Number of local news originators per municipality by percentage of population that is African-American.

| Weighted Count of Municipalities with Number of LNOs Serving Them | | Percent of Muni. pop. That Is African-American | | | Total |
|---|---|---|---|---|---|
| | | 0–3% of pop. Is Af. Amer. | 3–6% of pop. Is Af. Amer. | >6% of pop. Is Af. Amer. | |
| 0–2 LNOs per 10k pop. | Count | 37 | 10 | 40 | 87 |
| | % | 42.5% | 11.5% | 46.0% | 100.0% |
| 2–6 LNOs per 10k pop. | Count | 77 | 27 | 41 | 145 |
| | % | 53.1% | 18.6% | 28.3% | 100.0% |
| 6–2 LNOs per 10k pop. | Count | 67 | 21 | 33 | 121 |
| | % | 55.4% | 17.4% | 27.3% | 100.0% |
| 12–40 LNOs per 10k pop. | Count | 67 | 28 | 49 | 144 |
| | % | 46.5% | 19.4% | 34.0% | 100.0% |
| 40 or more LNOs per 10k pop. | Count | 41 | 10 | 17 | 68 |
| | % | 60.3% | 14.7% | 25.0% | 100.0% |
| Total | Count | 289 | 96 | 180 | 565 |
| | % | 51.2% | 17.0% | 31.9% | 100.0% |

Table 6 shows the number of local news originators serving a municipality as it relates to a community's Hispanic population. This structural measure is highly significant at the $p < 0.001$ level (see Appendix C), which echoes the findings in other studies (e.g., Napoli et al. 2018). The analysis shows that municipalities with the greatest percentage of Hispanic residents are the most likely to be news deserts, while the likelihood of having a

higher number of local news originators increases as the percentage of the population that is Hispanic decreases.

Similarly, Table 7 shows the number of local news originators serving a municipality as it relates to a community's African-American population. This measure is highly suggestive but not statistically significant, at the $p < 0.08$ level (see Appendix C), and one sees a pattern similar to that for percentage of Hispanics, wherein municipalities that have the highest percentage of African-Americans are also the most likely to be news deserts.

In sum, the comparative analysis presented above, of 565 communities, found the median household income, character of a community (rural, urban, suburban), amount of municipal spending, and percentage of the population that is Hispanic to be statistically significant in relation to the number of local news originators that say they cover a community. Educational attainment and percentage of the population that is African-American were found not to be strongly related to level of coverage.

## 4. Discussion and Conclusions

How can knowledge about the structural correlates of local news provision be used to help solve the local journalism crisis? I propose that there are two structural measures, both found here to be statistically related to local news provision, that really matter: median household income, and municipal spending.

The fact that local news provision is positively correlated to median household income has huge implications for the business model of local journalism going forward. So many organizations are successfully cracking the code to audience revenue: NYU's (now defunct) Membership Puzzle, American Press Institute, and News Revenue Hub, to name just a few. Many local outlets that may have otherwise had to close are now being sustained by audiences who genuinely care about and value—and, crucially, are able to pay for—their content. The audience–revenue model has become one of the most hopeful and celebrated ways for local journalism to find sustainability. However, the findings of this study suggest that the audience–revenue model may not work in low-income communities, and that overreliance on it may recreate the inequities documented here, as residents who lack disposable income will be unable to sustain a news organization.

The findings further suggest the need to seriously consider other models to fund local news, such as giving individuals cash or tax credits toward news outlet subscriptions, payroll tax relief for journalists' salaries, and subsidies for small business advertising—all of which are currently being considered by Congress as part of the Local Journalism Sustainability Act, HR 7640 (Edmonds 2020). In addition, more philanthropic money could go toward operational costs for news outlets rather than toward projects, as much of it is now.

Second, municipal spending. "Democracy dies in darkness", as *The Washington Post* says, and public spending of taxpayers' money is one of the most important areas in which journalism shines a light. New Jersey municipalities spent USD 15,133,713,970 billion in 2016, the year in which the budget data were gathered[17]. Almost *half*—USD 7,438,217,276—was spent in municipalities that are covered by zero to two local news originators per capita. This is because seven of the eight communities with the highest municipal spending and the lowest number of local news originators covering them are the largest cities in New Jersey: Newark, Jersey City, Trenton, Paterson, Elizabeth, Atlantic City, and Camden (the eighth is Woodbridge Township). Note that the LNO measure used here does not mean there is zero coverage; it means that the level of coverage per capita is far less than in suburban areas (of course, the content analysis will provide crucial measures of quality, which may show the outlets that do cover these cities to be providing excellent content). It is toward this finding that journalism-support organizations and philanthropies could directly target money and effort: coverage of urban municipal spending.

Finally, this study showed that Hispanic communities, the largest minority population in New Jersey, is generally underserved. One important caveat to this finding is that many ethnic outlets' coverage areas were not mappable, and were therefore not included in

the analysis. A previous study of ethnic and community media in New Jersey (Stonbely and Advincula 2019) found 27 ethnic and community media outlets serving Hispanic communities (the largest number for any group). Of those 27, 17 are mapped and are therefore included in this analysis, and 10 are not. Therefore, while there is some local news provision to the Hispanic population, this remains an area in which additional attention and investment are warranted.

Methodologically, the main advancements offered here are, first, that mapping local news outlets by coverage area rather than where they are based provides greater detail and therefore accuracy; and second, a scaled yet granular comparative empirical analysis of the structural characteristics of communities that foster news deserts and news oases. Of course, a major caveat remains: whether the communities that an outlet says it covers are actually being covered; i.e., whether journalists are paying attention to municipal spending and education and other important local issues and writing about them regularly. This is what the next phase of this project, the content analysis, will seek to answer. In addition, even if such stories are being written and published for a community, is anyone paying attention? Do the intended readers actually consume the content and use it in a meaningful way? Audience studies are also planned as part of this larger project.

As a final thought, though we have no other states to compare it to, and knowing that New Jersey is one of the most densely populated states in the country, 779 outlets is a rather robust number, suggesting that the "doom and gloom" narrative that so often accompanies discussions of local journalism may in fact be more nuanced than many believe. While there are clearly many avenues for improvement, a detailed portrayal of one state's local news ecosystems shows that there are bright spots as well.

**Funding:** This research was funded in part by a grant from the Tow Center for Digital Journalism at Columbia University.

**Institutional Review Board Statement:** Not applicable.

**Informed Consent Statement:** Not applicable.

**Data Availability Statement:** Supporting data for this research may be found at https://newsecosystems.org/. Due to the proprietary nature of some of the data, a full list of news outlets is not available; however the sum of outlets is available either through the search subpage or the map subpage.

**Acknowledgments:** Special thanks to Jesse Holcomb, who was instrumental in the creation of the map on which this analysis is based, and to Ashley Stiemle, who was a research assistant on this project. Thanks to The Tow Center for Digital Journalism, in particular Emily Bell and Pete Brown, for funding this phase of the project; thanks to Zev Ross and Sandy Haaf, who built the online map; Matt Weber; CCM's Stefanie Murray and Joe Amditis; and Fiona Morgan, for thoughtful comments on a final draft.

**Conflicts of Interest:** The author declares no conflict of interest. Furthermore, the funder had no role in the design of the study; in the collection, analyses, or interpretation of data; in the writing of the manuscript; or in the decision to publish the results.

## Appendix A

Databases of local news providers used. Lists marked with an asterisk (*) we had to purchase; the rest are available online, free of charge (accessed 2019–2020).

National

- Editor & Publisher Yearbook*
- Cision*
- BIA/Kelsey*
- Library of Congress
- Online Newspaper Directory
- Michele's List
- ABYZ
- National Newspaper Association

- Local Independent Online News Publishers (LION)
- Institute for Nonprofit News (INN)

  State-level

- New Jersey Press Association
- Proprietary to Center for Cooperative Media
- New Jersey News Commons
- Ethnic & community media serving NJ

**Appendix B**

Critical Information Needs (Friedland et al. 2012)

The descriptions have been modified to fit the journalism context; in the original, they more broadly address information environments.

1.   Emergencies and risks

News about policing and public safety, as well as information about emergencies such as dangerous weather; environmental and other biohazardous outbreaks; and public safety threats, including terrorism, Amber Alerts, and other threats to public order and safety.

2.   Public health

News about local health and healthcare, including information on family and public health; information on the availability, quality, and cost of local health care; the availability of local public health information, programs, and services, including wellness care and local clinics and hospitals; information on the spread of disease and vaccination; information about local health campaigns and interventions. *Not* including personal health "news you can use", such as information about new pharmaceutical drugs, exercise, or diets.

3.   Education

News about all aspects of the local educational system, including the quality and administration of local school systems; information about educational opportunities, including school performance assessments, enrichment, tutoring, and afterschool care and programs; information about school alternatives, including charters; information about adult education, including language courses, job training, and GED programs, as well as local opportunities for higher education.

4.   Transportation Infrastructure and systems

News about local transportation, including information about essential transportation services including mass transit at the neighborhood, city, and regional levels; traffic and road conditions, including those related to weather and closings; and public debate about spending on transportation infrastructure.

5.   Environment and Planning

News about the local environment, as well as local environmental issues that may affect the quality of life; news about the quality of local and regional water and air, hazards, and longer term issues of sustainability; the distribution of actual and potential local environmental hazards, including toxic hazards and brownfields; natural resource development issues; and news about restoration of watersheds and habitat.

6.   Economic Development

News about local employment; job training and retraining, apprenticeship, and other sources of reskilling and advancement; information on small business opportunities, including startup assistance and capital resources; information on major economic development initiatives affecting all local levels.

7.   Civic Information

News about civic institutions, nonprofit organizations, and associations, including their services, accessibility, and opportunities for participation. This category also includes

libraries and community-based information services; cultural and arts information; recreational opportunities; nonprofit groups and associations; community-based social services and programs; and religious institutions and programs. Does *not* include press-release-based or advertorial articles for purely cultural activities.

8. Politics and Political Life

News about candidates for office and elected officials in all units of governance; also includes information on elected and voluntary neighborhood councils, school boards, city council and alder elections, city regions, and county elections. News/information on public meetings and issues, including outcomes; information on where and how to register to vote, including requirements for identification and absentee ballots; information on state-level issues where they impact local policy formation and decisions. Accountability and corruption news regarding local politicians.

**Appendix C**

**Table A1.** Chi-Square test for Table 2, number of local news originators by median household income.

|  | Value | df | Asymptotic Significance (2-Sided) |
|---|---|---|---|
| Pearson Chi-Square | 26.021 [a] | 16 | 0.054 |
| Likelihood Ratio | 25.349 | 16 | 0.064 |
| Linear-by-Linear Association | 12.528 | 1 | 0.000 |
| N of Valid Cases | 565 | | |

[a] 2 cells (8.0%) have an expected count of less than 5. The minimum expected count is 3.01.

**Table A2.** Chi-Square test for Table 3, number of local news originators by educational attainment.

|  | Value | df | Asymptotic Significance (2-Sided) |
|---|---|---|---|
| Pearson Chi-Square | 20.432 [a] | 12 | 0.059 |
| Likelihood Ratio | 20.052 | 12 | 0.066 |
| Linear-by-Linear Association | 6.021 | 1 | 0.014 |
| N of Valid Cases | 565 | | |

[a] 2 cells (10.0%) have an expected count of less than 5. The minimum expected count is 3.61.

**Table A3.** Chi-Square test for Table 4, number of local news originators by character of community (population density).

|  | Value | df | Asymptotic Significance (2-Sided) |
|---|---|---|---|
| Pearson Chi-Square | 103.815 [a] | 8 | 0.000 |
| Likelihood Ratio | 107.167 | 8 | 0.000 |
| N of Valid Cases | 565 | | |

[a] 1 cell (6.7%) has an expected count of less than 5. The minimum expected count is 2.10.

**Table A4.** Chi-Square test for Table 5, number of local news originators by municipal spending.

|  | Value | df | Asymptotic Significance (2-Sided) |
|---|---|---|---|
| Pearson Chi-Square | 75.980 [a] | 16 | 0.000 |
| Likelihood Ratio | 68.096 | 16 | 0.000 |
| Linear-by-Linear Association | 22.758 | 1 | 0.000 |
| N of Valid Cases | 559 | | |

[a] 0 cells (0.0%) have an expected count of less than 5. The minimum expected count is 6.16.

**Table A5.** Chi-Square test for Table 6, number of local news originators by percentage Hispanic.

| | Value | df | Asymptotic Significance (2-Sided) |
|---|---|---|---|
| Pearson Chi-Square | 32.216 [a] | 8 | 0.000 |
| Likelihood Ratio | 32.298 | 8 | 0.000 |
| Linear-by-Linear Association | 9.843 | 1 | 0.002 |
| N of Valid Cases | 565 | | |

[a] 0 cells (0.0%) have an expected count of less than 5. The minimum expected count is 21.78.

**Table A6.** Chi-Square test for Table 7, number of local news originators by percentage African-American.

| | Value | df | Asymptotic Significance (2-Sided) |
|---|---|---|---|
| Pearson Chi-Square | 14.052 [a] | 8 | 0.080 |
| Likelihood Ratio | 13.735 | 8 | 0.089 |
| Linear-by-Linear Association | 2.632 | 1 | 0.105 |
| N of Valid Cases | 565 | | |

[a] 0 cells (0.0%) have expected an count of less than 5. The minimum expected count is 11.55.

## Notes

[1]  Access to reliable broadband internet is a related issue that is not studied here (e.g., Reddick et al. 2020).

[2]  Each municipality's population (for the year 2017, retrieved from the U.S. Census Bureau website (https://www.census.gov/) (accessed on 11 February 2023), was divided by the municipality's geographic size in square miles (from NJGIN Open Data: https://njogis-newjersey.opendata.arcgis.com/) (accessed on 11 February 2023), then grouped into the categories rural, suburban, and urban, where rural is ≤500 people per square mile, suburban is 501–9000 people per square mile, and urban is >9000 people per square mile (see Ratcliffe et al. 2016).

[3]  It is also the case that municipalities with larger budgets should, in theory, generate a greater number of public notices, which are required by law to be advertised in a newspaper. The proceeds from public notices are one of the main and only ways that the United States currently provides public funding to media (Waldman 2011, pp. 334–35).

[4]  In 2022, just after this research was concluded, New Jersey combined two of its municipalities for a new total of 564 municipalities.

[5]  For example, both Montclair and Newark are located in Essex County, NJ. Montclair is an affluent, highly educated town (home to many journalists who are based in New York City), while Newark has a much lower median household income and has an education system that has been the subject of highly publicized philanthropic interventions. To discuss local news provision for Essex County at the county level obscures all of these important differences.

[6]  For a good overview and description of this more inclusive definition, see Thorson et al. (2020).

[7]  Median household income data used here are from the years 2016 or 2017 and come from census.gov/quickfacts.

[8]  Educational attainment data are from the year 2019 and come from factfinder.census.gov; the number I use here is the sum for each municipality of the percentage of adults age 25 and older with a bachelor's degree plus the percentage of adults age 25 and older with a graduate or professional degree.

[9]  Municipal spending data are from the years 2017 or 2018 and come from the State of New Jersey Department of Community Affairs (https://nj.gov/dca/divisions/dlgs/resources/fiscal_rpts.shtml) (accessed on 11 February 2023).

[10]  Percentages of Hispanic and African American residents in each municipality are from 2019 and come from New Jersey Data Book, run by Rutgers Center for Government Services (https://njdatabook.rutgers.edu/) (accessed on 11 February 2023).

[11]  To gut-check the television and radio numbers we consulted Bob Papper, a distinguished educator, researcher, and journalist with more than 40 years of experience (www.bobpapper.com) (accessed on 11 February 2023), who has conducted research for the Radio Television Digital News Association (RTDNA) since 1995 and is currently director of the RTDNA/Newhouse School at Syracuse University Annual Survey. He thought our radio numbers seemed a little low, until I explained that we included only those that produce local news; he thought our television numbers seemed high—he would've guessed there were 40 to 50 stations serving New Jersey rather than 100 (Papper 2020). However, our television number includes 58 municipal-access stations, and after subtracting those we have 42 local television stations, squarely within his window. The analysis below includes those 58 stations because they do likely provide the kind of local news that we are interested in, regardless of how many or how few viewers they have. The next phase of this research project is a content analysis that will look at, among other things, which towns are covered by each outlet, lending further clarity to this issue.

12     This standard for analyzing content—of looking at localism, originality, and coverage of a critical information need—was developed by Philip Napoli (see e.g., Napoli et al. 2018). The full list of critical information needs can be found in Appendix B.

13     The careful reader will note that 565 is also the number of municipalities in New Jersey, a confusing coincidence.

14     Note that the weighted numbers of local news originators per municipality in all tables below are higher than the actual number of LNOs.

15     The 10 New Jersey municipalities with the largest Hispanic populations are Newark, Paterson, Passaic City, Jersey City, North Bergen Township, West New York Township, Union City, Elizabeth, Perth Amboy, and Camden (Wu 2011).

16     The 10 municipalities with the largest African American populations are Newark, Jersey City, East Orange, Irvington Township, Paterson, Trenton, Camden, Elizabeth, Plainfield, and Willingboro Township.

17     The budget data were gathered early on in the project. A spot-check of more recent figures showed that they have remained very similar in recent years.

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
