# Peer review of "What Makes for Robust Local News Provision? Structural Correlates of Local News Coverage for an Entire U.S. State, and Mapping Local News Using a New Method"

_journalmedia, doi:10.3390/journalmedia4020031_

Round 1

Reviewer 1 Report

Overall, this is a very valuable addition to the literature on the challenges of local news provision and the methodological problems of building frameworks to audit and assess the adequacy of local journalism in complex geographies.

The only suggestions I would make are largely stylistic and relate to the Discussion/Conclusion. The incorporation of numerical listing (1. ... 2. ... - lines 603-606) within the text doesn't match the presentation in the rest of the writing and could be expressed in prose instead.

In addition, the caveat relating to coverage of ethnic media due to the mapping problem (lines 595-600) could be expressed more clearly earlier in the paper. It is mentioned in lines 419-423 and the nature of the methodological challenge is adequately explained, but perhaps the implications for the subsequent results could be made more explicit at the earlier mention (e.g line 420).

Author Response

Reviewer revision: The incorporation of numerical listing (1. ... 2. ... - lines 603-606) within the text doesn't match the presentation in the rest of the writing and could be expressed in prose instead.

Response: I have taken out the numbers and written it as text (now lines 607-609).

Reviewer revision: In addition, the caveat relating to coverage of ethnic media due to the mapping problem (lines 595-600) could be expressed more clearly earlier in the paper. It is mentioned in lines 419-423 and the nature of the methodological challenge is adequately explained, but perhaps the implications for the subsequent results could be made more explicit at the earlier mention (e.g line 420).

Response: I have added lines 427-430 emphasizing that many ethnic outlets are not included in the analysis because they are not mappable according to this method.

Reviewer 2 Report

The text offers insightful findings on the main significant indicators of local news provision. The approach is original, showing relevant correlations. On this matter, the concept of “local news originators” works well; hence, the map provided could be developed in additional contexts.

Regarding the sample, it is small, but interesting as a case-study. In this sense, it is necessary to know why New Jersey seems relevant and which reasons lead to select the newspapers.

The list of references is appropriate and up-to-date, mixing classic and recent books and articles.

This is a well written and really interesting article. In my opinion, minor revisions are needed prior to publication both in the introduction and the conclusions. The authors might want to consider adding proper objectives, research questions or hypotheses in the introduction, which would make easier to follow the paper.

Moreover, I miss some references at the end of the study. Median household income and population density are variables that were present on some prior scholarship on local media. Therefore, it would be useful to situate the current contributions in the framework of wider debates.

Author Response

Reviewer revision: Regarding the sample, it is small, but interesting as a case-study. In this sense, it is necessary to know why New Jersey seems relevant and which reasons lead to select the newspapers.

Response: If I include this text during the blind review it may give away the author, so I suggest it be included after the review is finished; I would insert it as a footnote at the end of the sentence that reads, “Second, I identify, based on these coverage areas, the structural correlates of local news deserts for an entire U.S. state (New Jersey) and in doing so contribute to the comparative empirical question of which community characteristics are most closely related to local news provision” (line 63-66):

The state of New Jersey was chosen because it has been one of a handful of states that has acted over the last several years as a laboratory for philanthropic funding of local news (de Aguiar and Stearns, 2016). One of the results is that the state is home to a network of local news outlets called New Jersey News Commons, housed at the Center for Cooperative Media, where the author serves as research director.

*need to add citation to references:

de Aguiar, M. and Stearns, J. (2016). New Report: Lessons Learned from the Local News Lab. Geraldine R. Dodge Foundation. Accessed at https://localnewslab1.wpengine.com/2016/02/19/new-report-lessons-learned-from-the-local-news-lab/

Reviewer revision: The authors might want to consider adding proper objectives, research questions or hypotheses in the introduction, which would make easier to follow the paper.

Response: Added the sentence, “Each structural feature currently hypothesized to relate to local news provision is discussed below.” (line 73-74)

Reviewer revision: Moreover, I miss some references at the end of the study. Median household income and population density are variables that were present on some prior scholarship on local media. Therefore, it would be useful to situate the current contributions in the framework of wider debates.

Response: This reviewer also says, earlier, “The list of references is appropriate and up-to-date, mixing classic and recent books and articles.”

If there are specific references the reviewer thinks are critical to add, it would be super helpful to have them stated explicitly. As it is, I made a point of being extremely thorough in citing existing relevant literature.